# The Effects of Cognitive Task and Change of Height on Postural Stability and Cardiovascular Stress in Workers Working at Height

**DOI:** 10.3390/ijerph17186541

**Published:** 2020-09-08

**Authors:** Magdalena Cyma-Wejchenig, Janusz Maciaszek, Katarzyna Marciniak, Rafał Stemplewski

**Affiliations:** Department of Physical Activity and Health Promotion Science, Poznan University of Physical Education, 61-871 Poznan, Poland; jmaciaszek@awf.poznan.pl (J.M.); katarzyna.anna.m@gmail.com (K.M.); stemplewski@awf.poznan.pl (R.S.)

**Keywords:** postural stability, cognitive task, at-height workers, cardiovascular stress

## Abstract

The purpose of the study was to analyze the effects of cognitive task and change of height on the postural stability and cardiovascular stress of at-height workers. The study included 32 healthy men aged 25–47. Due to the type of work performed, two groups were identified: at‒height workers, HW (*n* = 16), and office workers (mainly work at desk with a computer) OW (*n* = 16). The objective measures of postural stability (posturography) and cardiovascular stress (heart rate monitor) were evaluated for both groups at two different platform heights (ground level and 1 m above the ground) with or without cognitive task (backward counting). The increased height and the cognitive task were found to significantly affect measures of postural stability and cardiovascular stress. It was observed that in inexperienced OW employees, higher platform height and performing a cognitive task meant that posture stability significantly decreased, while cardiovascular stress and difficulties in maintaining balance increased. In HW group postural stability is less affected by distress conditions than in OW group.

## 1. Introduction

The construction industry and in particular work at height are often considered to be one of the most dangerous occupations [1]. According to the Occupational Safety and Health Administration [2], any work during which the distance between the working platform and the ground involves a fall from one level to a lower level is work at height, e.g. on scaffoldings or building structures. The “Labor Code” [3] recommends examinations of the balance system of people working at above one meter from ground/floor level. This type of work may result in an accident or illness [2,4,5]. Every year in the UK, 10 million people perform different forms of work at height and falls from a height cause nearly three out of ten fatal injuries (29%) [6,7]. In 2013/2014 alone, they caused losses estimated at 567,000 working days: that is why understanding all the possible challenges related to working at height becomes necessary [4,6]. In the European Union countries, one of the most common causes of accidents during work at height is low risk awareness. This unawareness leads to downplaying the hazardous behaviors, which is associated with employees’ lack of knowledge of possible dangers and of suitable procedures in the event of an accident [1]. Only highly qualified and experienced workers with the appropriate mental and physical qualifications are competent to work at height. People working at height should undergo medical examinations, including ophthalmological, neurological and electrophysiological tests [8]. Li and Poon [9] reported that the causes of accidents at height can be classified as technical, organizational, and human.

Among the most common causes of accidents related to work at height are postural stability deficiencies [10]. When working at height, unsafe environmental conditions, inappropriate safety equipment or lack thereof, insufficient collective protection measures, uneven work surfaces, slippery surfaces and the presence of obstacles can lead to loss of postural control [11]. Falls occur when employees suddenly lose their balance as a consequence of slipping, tripping or spraining an ankle [5]. The reason for this situation may be a disorder of one of the functions associated with maintaining proper control of body posture [12,13,14,15]. A number of studies have been conducted that compared and assessed the risk of falls in view of physiological and psychological factors [16,17,18,19,20].

It has been identified that increasing the elevation of the surface on which the worker stands alters postural control [21,22,23]. While maintaining an upright stance, workers were observed to use a strategy of stiffening the body, characterized by reduced amplitude and increased frequency of posture adjustment as well as leaning backwards away from the direction of danger [14,24]. Modifications of postural control are accompanied by psychological [25], and physiological changes because participants are less confident, more afraid of falling, are more anxious and feel less stable when standing on an elevated platform [19,26,27]. It has been confirmed that workers are able to consciously intervene in postural control by increasing ankle stiffness in an upright stance [28,29]. It is important to identify whether postural threat results in a more conscious postural control and whether such a change in cognitive strategy can be linked to changes in postural control [21,30,31].

Min et al. [31] investigated whether safety handrails and scaffolding height affect the subjective and objective assessment of postural stability, and whether they affect cardiovascular stress, in novice and experienced construction workers. They reported that by increasing the altitude of the surface on which the studied person stands, the risk of fall perception rises. This applies especially to people who do not work at heights on a daily basis. The same authors also identified that the perception of loss of balance is reflected in the control of body balance as it alters the strategy adopted by the central nervous system [31]. Chander et al. [32] analyzed the impact of Virtual Reality (VR) generated construction environments at different heights on postural stability and fall risk. Studies have shown that the VR environment, regardless of virtual growth of the elevation platform, also induced increased postural instability. It has been suggested that this can be attributed to visual sensory conflicts in the postural control system created by the VR exposure.

Other studies have confirmed that the effect of elevated height and lack of safety handrails increases anxiety and subjective stress triggered by fear of falling, which has been demonstrated to affect the neuromuscular system in both healthy and unhealthy people [19,20,28]. Similar reactions were also noticed in a group of at-height workers [13,17,23,31]. Respondents often use more conservative strategies to maintain postural control in settings of increased hazard arising from, e.g., elevation change [21,33].

Zaback et al. [25] analyzed the adjustment of the parameters of emotional state and balance in their subjects after repeated exposure to postural threat caused by an increase in height. The emotional response of individuals was weakened after repeated exposure to danger. However, the change in standing balance did not change significantly despite exposure to danger. The results may suggest that some threat-induced balance changes are more closely allied with an emotional response than others. Johnson et al. [34], who researched emotional, cognitive and postural adaptations in young and older adults under the influence of repeated exposure to danger, noticed that some modifications in threat-related postural control may be closely correlated with emotional response to the threat, while others may be dependent on the context. The emotional context was also emphasized in other research concerning human postural control [30,35,36].

Another factor in different perceptions of the threat to postural stability may be the performance of a dual task, or task switching, which is used in research on the engagement of attention in postural control [37]. In these studies, maintaining body posture is considered a basic task and concurrently performing a dual task is the secondary task [38]. Cullen and Agnew [39] have reported that performing a dual task can decrease the effectiveness of postural control compared to performing a single task. In addition, appropriate reweighting of attention is very important to prevent loss of balance when performing a double task [37]. It has been reported especially that recovering balance is very demanding in terms of information processing [40]. When insufficient attention resources allocated to postural tasks are processed, the risk of losing balance and consequently falling rises [41]. In our previous study [42], we analyzed the postural stability and physical activity of workers working at height. To assess postural stability, the one-leg standing test with eyes open and closed was used. The HW group obtained higher results in the postural stability functional test. Based on the results, it could be assumed that postural stability is influenced by exposure to stressful conditions such as working at height.

Despite the fact that the impact of threat to posture on modification of balance control has been previously documented, there is still little information about the mechanisms and strategies responsible for these changes, especially in the group of at-height workers [43,44,45]. One of the possible strategies may be to link the threat to more conscious control of body balance [21]. It is suggested that at-height workers are characterized by the increased automation of the postural stability system [46,47]. Previous studies have focused mainly on the effect of specified conditions (cognitive single-double task) or a specified level of height (low-high threat). To our best knowledge, there have been no studies that have examined all these elements together in the group of at-height workers.

The aim of this study was to assess differences in postural stability under various conditions: both when changing the height of the measuring platform and with an additional cognitive task. Changes in heart rate were evaluated in dangerous conditions caused by the change of the height platform. Hypotheses were as follows: (1) increasing the altitude of the measuring platform disturbs postural stability to a greater extent in the group of office workers than in high-altitude workers; (2) postural stability while performing a cognitive task is higher in the group of high-altitude workers than among office workers; (3) when changing altitude, greater cardiovascular stress occurs in office workers than in high-altitude workers.

## 2. Materials and Methods

### 2.1. Characteristics of the Study Group

The study involved 16 healthy men working at height (HW: high-altitude workers). 16 office workers were examined as the control group (OW: office workers, mainly working at desk with a computer). Eligibility criteria were as follows: minimum age of 25, verbal communication skills enabling informed, logical answers, full mobility, and a minimum of one year’s experience of working at height for the HW group.

All men interested in participating in the experiment agreed in writing to the experimental procedure and were informed in detail about the study procedures (participation was voluntary). The study was approved by the Bioethics Committee of the Poznan University of Medical Sciences (Decision No. 1111/16) and was in line with the Helsinki Declaration [48]. The basic characteristics of both groups were examined before the experiment. No statistically significant differences were identified between the groups in terms of age, BMI, and physical activity (PA) (Table 1).

### 2.2. Procedures

The research procedure included performing posturography tests (in random order to avoid the learning factor) which assess human body balance on a stabilometric platform at ground level and at a height of one meter from the ground (Figure 1).

Each trial test was made twice and lasted 30 s. In previous studies, it was identified that averaging two results is sufficient to obtain an ICC reliability coefficient above 0.9 for the average velocity of center or pressure (COP) [49]. It has been previously reported that recording COP movements for 30 s during a static posture is appropriate to record a reliable measurement [50]. 20-s intervals between the measurements were used (due to the large number of trials). Participants were able to rest in a sitting position in case of dizziness or tiredness. None of the participants took advantage of this opportunity, therefore during all the tests at the low or high level they stayed on the same level of the platform.

A stable, one-meter-high pedestal was used to place the stabilometry platform at a height. To ensure safety, gymnastic mattresses were placed around the platform (Figure 2). Tests were performed on the ground and at height with two different tasks:
Quiet standing with one’s eyes open: the subject stood still in the center of the platform with bare feet hip-width apart and arms down at sides. The posturography platform was placed three meters in front of a white wall, which subjects were asked to look directly at.Cognitive standing: the subject stood freely in the above-mentioned position and additionally performed a mathematical task, which consisted of counting backwards every third number down from the number 200 during the time of recording the data [37].

Participants’ heart rate was measured during tests in all the experimental conditions.

### 2.3. Measurement

#### 2.3.1. Level of Physical Activity (PA)

Some studies reported that PA can modulate the posture control of people of all ages [51,52]. The level of PA was assessed to exclude its impact on the posture control level of both groups.

The level of PA was assessed using the Caltrac activity monitor (Muscle Dynamics, Inc., Tarrance, CA, USA). The accelerometer produces an outcome based on the weekly measurement of energy expenditure due to PA [53,54]. Subjects wore the Caltrac for seven days. The total results in kilocalories were divided by the number of days.

#### 2.3.2. Postural Stability (PS)

COP data was collected using an AccuGait portable force plate (AMTI PJB-101 model, AMTI, Watertown, MA, USA). The plate was connected to the computer using the Balance Trainer software provided by the manufacturer. The sampling frequency was 100 Hz. The fourth-order lowpass Chebyshev II filter [55] with 10-Hz cutoff frequency [56] was used to filter raw data of COP signals. The length of the sway path (SP) of the COP signal and its components in the anteroposterior (AP) and mediolateral (ML) directions were analyzed. This is commonly used as a PS indicator [49,57].

#### 2.3.3. Cardiovascular Stress

The increase in heart rate on exercise compared to resting heart rate was measured as a physiological and psychological indicator of stress. Heart rate was measured for the first minute while the participants were executing the task at low and high threat [31,58,59]. The heart rate was measured with an automatic blood pressure monitor (Oro-Med, ORO-SM2 COMFORT, Warsaw, Poland) placed at the wrist. The pressure gauge has a special sensor for reading blood pressure and change in heart rate. The pressure signal from the blood vessel is read by a pressure sensor. The signal is later amplified and filtered to separate the heartbeat signals [58,60].

#### 2.3.4. Statistical analysis

Statistical analyzes were calculated using STATISTICA Software 13 (TIBCO Software Inc., Palo Alto, CA, USA). Statistical significance level was defined at *p* ≤ 0.05. The differences between the groups with respect to basic characteristics (age, BMI, physical activity indicators) were calculated using the t-test. The level of postural stability and cardiovascular stress was analyzed using two-way analysis of variance (ANOVA). The following interaction effects were evaluated: “height × group” and “task × group” (with two levels for each factor: low-high threat and with or without cognitive task among the two groups: HW and OW, respectively), the main effects for the study conditions (“height”, “task”) and the inter-group effect (“group”).

## 3. Results

### 3.1. Influence of Height on the Postural Stability of Employees

There was no interaction effect (“height × group”) for the sway path (SP), the anterior-posterior sway path (SPAP) or the mediolateral sway path (SPML) in conditions of quiet standing (F = 1.03, *p* > 0.05, η^2^ = 0.03; F = 0.78, *p* > 0.05, η^2^ = 0.02 and F = 1.04, *p* > 0.05, η^2^ = 0.03, respectively). Regardless of the conditions under which the measurement was carried out in the OW group, statistically significant higher path length measurements for SPML were identified – the intergroup effect (F = 7.37, *p* < 0.01, η^2^ = 0.20)—which may indicate a lower level of postural stability. A significant main effect “height” was observed for SP and SPAP (F = 7.16, *p* < 0.05; η^2^ = 0.19 and F = 10.59, *p* < 0.01, η^2^ = 0.26, respectively). This indicates that the change in altitude results in a lower level of the outcomes in both groups.

No significant effect of “height × group” interaction was identified in the cognitive standing for any of the tested parameters SP, SPAP and SPML (F = 0.09, *p* > 0.05, η^2^ = 0.00 F = 1.02, *p* > 0.05, η^2^ = 0.03 and F = 0.86, *p* > 0.05, η^2^ = 0.02, respectively). Regardless of the change in the conditions, higher results were identified for SP and SPML (F = 6.63, *p* < 0.01, η^2^ = 0.18 and F = 12.67, *p* < 0.001, η^2^ = 0.30, respectively) in the OW group (Figure 3).

### 3.2. The Impact of Performing a Cognitive Task on Employees’ Postural Stability

No “task × group” interaction effect for SP, SPAP or SPML is: (F = 0.002, *p* > 0.05, η^2^ = 0.00; F = 0.91, *p* > 0.05, η^2^ = 0.03 and F =1.04, *p* > 0.05, η^2^ = 0.03, respectively) when switching tasks during measurement at ground level. Regardless of the conditions in which the measurement was carried out, the HW group obtained statistically significant lower values for SP and SPML—the intergroup effect (F = 5.38, *p* < 0.05, η^2^ = 0.15 and F = 11.87, *p* < 0.01, η^2^ = 0.28, respectively)—which may indicate a higher level of postural stability. A significant main effect (“task”) was identified for SP and SPAP (F = 5.17, *p* < 0.05, η^2^ = 0.15 and F = 7.46, *p* < 0.01, η^2^ = 0.20, respectively). This suggests that performing the cognitive task causes deterioration of results in both groups.

No significant effect of the “task × group” interaction was noted at the increased height of the measuring platform during the performance of the task for any of the tested parameters SP, SPAP and SPML (F = 1.99, *p* > 0.05, η^2^ = 0.06; F = 1.13, *p* > 0.05, η^2^ = 0.04 and F = 2.04, *p* > 0.05, η^2^ = 0.06, respectively). Regardless of the change in the conditions, the OW group obtained higher path length values for SPML (F = 8.31, *p* < 0.01, η^2^ = 0.22). No statistically significant main effect of “task” was identified for any of the tested parameters (F = 2.07, *p* > 0.05, η^2^ = 0.06; F = 1.98, *p* > 0.05, η^2^ = 0.06 and F = 0.22, *p* > 0.05, η^2^ = 0.01, respectively). Regardless of the group and height of the measuring position, higher results were recorded during cognitive standing (Figure 4).

### 3.3. The Influence of Altitude on Cardiovascular Stress

Analysis of cardiovascular stress reported a significant interaction effect (F = 11.25, *p* < 0.01, η^2^ = 0.27). No statistically significant intergroup effect was reported (F = 0.21, *p* > 0.05, η^2^ = 0.01). However, at the height of 1 m, the OW group was characterized by higher values of heart rate. A statistically significant main effect, “height,” was noted (F = 84.35, *p* < 0.001, η^2^ = 0.74). Regardless of the group, higher results were recorded at the increased height of the measuring platform (Figure 5).

## 4. Discussion

To our best knowledge there are no previous studies comparing HW group with OW group. However, the overall view of the results, related to changes in stability with height of the platform and cognitive tasks, is consistent with other studies where at height workers were examined in other contexts.

The obtained results confirm that the change in height causes a deterioration in a quiet standing stance regardless of the studied group (HW and OW) [26,28,45,61]. In quiet standing tests, both increase in the length of the COP path and range of sway were observed along with the increase in height at which the subject was standing. For all the analyzed parameters lower results (which may indicate a higher level of postural stability) were achieved by the HW group. Presumably, higher posture stability in HW group can be explained by the daily training of individual muscle groups used in the professional tasks of high-altitude workers, from balance training related to the conditions and nature of their work, as well as from their overall level of physical activity performed at work [30,43,62].

Adkin et al. [28] analyzed changes in posture control at different heights of the measurement platform above ground level. The change in platform height has also been used to modify threat levels. It has been reported that the control of postural stability is closely affected by threat to standing position, but also by the order in which the postural threat occurred. In addition, the control of postural stability increases with level of experience (i.e. previous experience of postural threat). Similar results were reported by Sturnieks et al. [26] who tested the effect of age, anxiety and fear of falling on postural sways in young and older adults standing on a 65-cm-high platform. In response to postural threat, people experiencing anxiety adopted strategies for improved balance control by increasing the frequency of COP sways and minimizing sway ranges. It has been also identified that the 65 cm height used in the study is not adequate enough to induce changes in balance control in a group of young adults.

The findings for the cognitive standing sample revealed a higher length of the COP path as compared to quiet standing. Height did not change the subjects’ balance control despite significant differences in COP signal parameters between quiet standing and cognitive standing. A much lower level of changes in COP and its components towards AP and ML planes was noted in the HW group during cognitive standing and after the change in height. These results may suggest that carrying out the task and changing the height do not significantly influence the level of postural stability in the group of experienced HW [39,40,63]. Fear of falling and previous experience of work at height can play an important role in altering posture control strategy in conditions of high risk due to altitude change [35,64,65]. It has been confirmed that the mode of the sensory systems involved in postural control is shaped by different levels of anxiety depending on the individual’s intensity of fear [44,66]. In addition, when performing tasks or processing visual information, people who are less resistant to fear and to stressful situations may use strategies which are less conducive to maintaining postural control in conditions of increased postural threat, instigated, for example, by increasing the height of the platform [65,66,67,68]. As a consequence, this may lead to greater postural instability in these people [65,66,69].

Pellecchia [70] investigated whether postural balance changes under the influence of three cognitive tasks depending on their difficulty. Young adults participated in the study. The results indicated that performing the most difficult task, i.e. the counting backwards using every third number. impacted postural sway. The longest COP path was recorded for the sample which involved counting every third number backwards. The path length increased with the difficulty of the task. It may confirm the results of this study.

Hainaut et al. [66] analyzed the effect of moderate anxiety on static balance with open and closed eyes in two groups of healthy people with contrasting features of anxiety. It has been reported that anxiety induces a larger and faster body sway in both groups in the open eye test. This suggests that anxiety may modify the processing of various sensory data involved in balance control irrespective of the level of anxiety of the subjects. Results regarding inter-individual differences point to a mutual relationship between static balance control and anxiety.

Based on this research, it can be concluded that typical tasks performed during work at height, as well as increasing awareness of the postural threat in HW, contribute to a slight level of change in postural stability, regardless of the industry and age of the employee [63,64,71]. It is also possible that the high-risk state in the OW group begins at a lower elevation than one meter, which causes high individual variability in level of postural stability [19,26].

Cardiovascular stress is a psychological and physiological indicator that increases with growing height and danger [16,31,60]. In the OW group, a higher surge in cardiovascular stress was observed with a rise in perceived difficulty in maintaining balance due to height and task performance [18,31,72]. The HW Group experienced a lower level of mental stress due to the increase in platform height [31,73]. Lower levels of cardiovascular stress may be associated with professional experience related to tasks such as building, dismantling and modification of scaffolding carried out on tall buildings that force employees to handle both precise work and difficult weather conditions, such as strong wind, rain and snow [27,63]. It is possible that the change in platform height from low to high (one meter) may not be dangerous enough to cause sufficient individual variability in the HW group [19,26].

Similar conclusions were revealed by the research of Hsu et al. [73] who examined the effect of altitude changes on mental stress in workers building high voltage transmission towers. Twelve experienced male employees aged 29 to 58 participated in the experiment. The study analyzed the impact of working surface elevation on heart rate variability of employees during the construction of a high voltage transmission tower on a distant and exposed mountain slope at three different heights. The results of the study demonstrate that the height of the working surface significantly affects heart rate variability (HRV). Construction employees working on high voltage transmission towers exhibit an increased level of mental stress as the elevation of the working platform increases, which may be attributable to insecurity, lack of safety guards, work environment and visual perception. Min et al. [31] also reported a significant impact of work experience, scaffolding height and the presence of safety guards (handrails) on the level of postural stability and cardiovascular stress. With lower employee experience, a higher scaffolding height and absence of security guards, postural stability was significantly reduced, while cardiovascular stress grew. These results were supported by the findings of this study.

Finally, it should be mentioned that this work is limited. The tests were performed only at ground level and at a height of one meter. This height may not have been adequate to produce sufficient individual variation in both groups. Increasing the platform height can significantly increase the difference between groups. An accurate assessment of the emotional state of the subjects could help improve the analysis of the obtained outcomes.

## 5. Conclusions

It was reported that COP path lengths, cardiovascular stress, and difficulties in maintaining balance during task performance are statistically significantly lower in experienced HW employees. This study provides evidence that professional experience and individual traits, including specific personal qualities and differences, can shape a strategy of postural control adopted in conditions caused by an increased postural threat. Although this research does not have a direct impact on improving the assessment and treatment of balance problems, it is suggested that it may be important in understanding how particular individual characteristics and professional experience affect the level of postural stability in difficult or dangerous conditions. Further studies of level of postural stability, taking into account psychological factors, fear of falling, and the impact of balance training on the above-mentioned aspects are warranted to reduce the risk of accidents among high-altitude workers.

## Figures and Tables

**Figure 1 ijerph-17-06541-f001:**
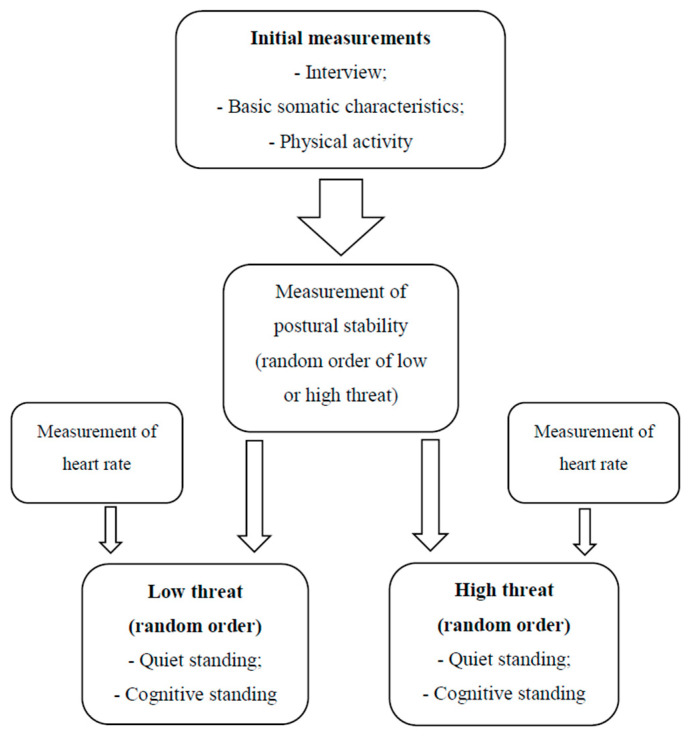
General overview of the experiment.

**Figure 2 ijerph-17-06541-f002:**
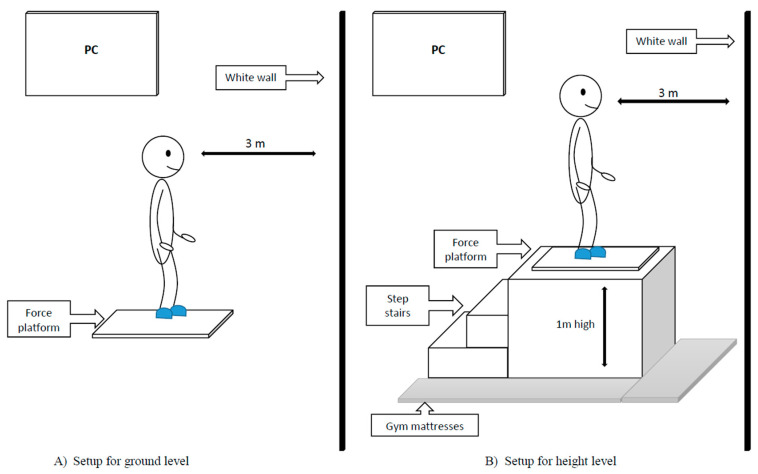
Experimental setup of two measuring stations at ground level (**A**) and at a height of 1 m from the ground (**B**).

**Figure 3 ijerph-17-06541-f003:**
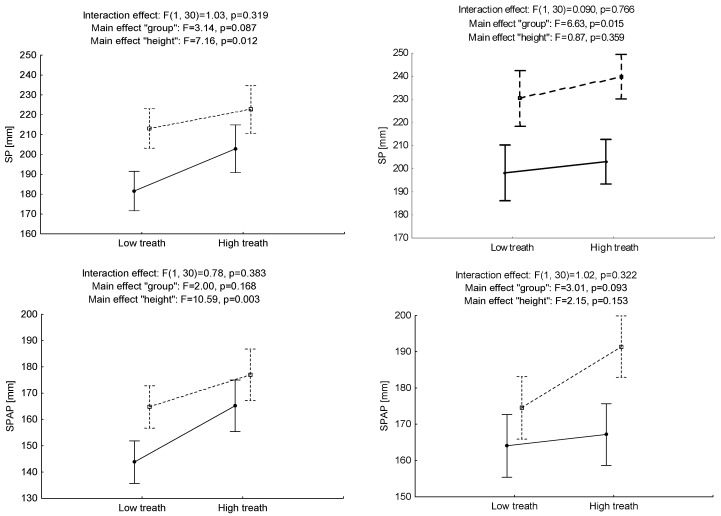
Mean values and standard error of measurements for sway path of center of pressure (COP) displacements (SP) and its components in anteroposterior (AP) and mediolateral (ML) directions for ‘‘height’’ factor (low-high threat) and ‘‘group’’ factor (at-height workers—HW and office workers—OW) for quiet standing (**A**) and cognitive standing (**B**).

**Figure 4 ijerph-17-06541-f004:**
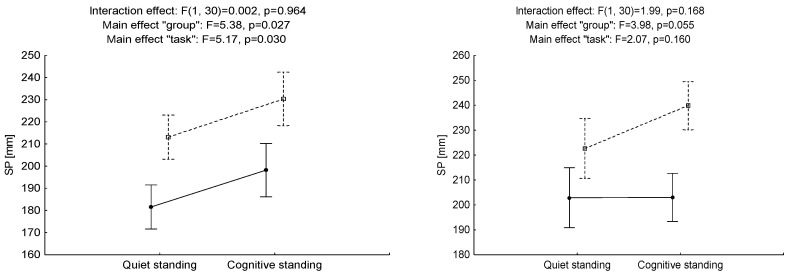
Mean values and standard error of measurements for sway path of COP displacement (SP) and its components in AP and ML directions for ‘‘task’’ factor (quiet standing and cognitive standing) and ‘‘group’’ factor (at-height workers—HW and office workers—OW) for low threat (**A**) and high threat (**B**).

**Figure 5 ijerph-17-06541-f005:**
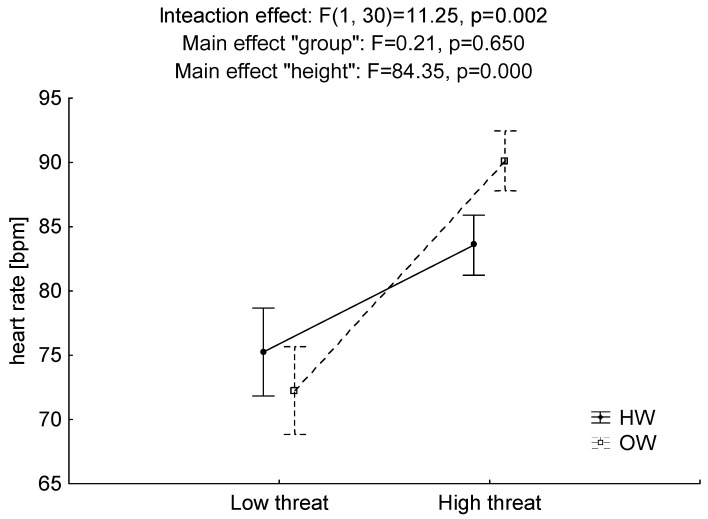
Mean values and standard error of measurements for cardiovascular stress at low and high postural threat in groups of at-height workers (HW) and office workers (OW).

**Table 1 ijerph-17-06541-t001:** Average values, standard deviations and differences between groups for the general characteristics of the participants and physical activity before the start of the experiment.

Variable	M (sd) HW	M (sd) OW	*t* df = 30	*p*
Age [years]	34.5(7.49)	36.00(6.31)	−0.61	0.54
Body height (m)	1.82(0.05)	1.79(0.08)	1.25	0.22
Body weight (kg)	90.5(9.94)	83.5(13.34)	1.69	0.23
BMI [kg/m2]	27.31(2.59)	26.01(3.41)	1.21	0.23
PA [cals used/wk]	21918.94 (2962.42)	19694.56 (4178.72)	1.74	0.09

Note. HW–height workers; OW–office workers; BMI–body mass index; PA–physical activity.

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
