# Peer review of "The Effects of Cognitive Task and Change of Height on Postural Stability and Cardiovascular Stress in Workers Working at Height"

_ijerph, 2020, doi:10.3390/ijerph17186541_

Round 1
Reviewer 1 Report
Dear Authors,
Thank you for your interesting, well written manuscript.
Your study is scientifically sound and well presented. Before I can recommend its publication in IJERPH, please address the following issues, though.
Major issue:
One meter of height above ground (line 123) does not seem to represent an adequate level of apparent threat as “falling” down this distance (on gymnastics mattresses) is unlikely to produce any kind of discomfort to the subjects. It is beyond question, of course, health and safety of participants must be guaranteed throughout any study. However, increasing the height to. e. g., 1.5 m or even 2 m, when adequately protected by mattresses, would have been equally harmless for healthy adult men. Results for those heights would have been more revealing for the outcome of the study, as you discuss yourselves in lines 298 and 314–317. So, why did you choose only 1 m of height?
Minor issues:
- Lines 103–104: “When changing altitude, greater cardiovascular stress occurs in office workers.” Please clarify if that “greater” refers to the intra-group (OW) or inter-group comparison. What was your corresponding assumption for HW?
- Line 109: What was your precise definition of OW? E.g., starting at what average numbers of hours seated per business day?
- Line 113: Please state whether or not your research was conducted in accordance with the declaration of Helsinki.
- Line 245: "Overall level of physical activity”: According to your group comparisons, PA as evaluated with your tracking device before your experimental trials was *comparable* for both groups. So how can you implicitly suggest that PA was higher for the HW group as compared to the OW group? – Even though this assumption does seem conceivable to me as well. However, you have to stick to your data, or explain why your data may not be sufficient to provide a complete picture.
Language issues:
- Line 126: “Each trial *lasted* 30 seconds…” (erase “was”).
- Line 129: is *an* appropriate time (replace “the”)
- Line 133: “They kept…” (erase left)
- Lines 161, 182 and thereafter: mediolateral (replace “medial-lateral”)
- Line 164: “for *the* first minute” (erase one)
- Line: 181 “It has been reported…”: Replace by “There was no…”
- Line 183 and thereafter: Please set eta^2 in formulae style (2 as superscript)
- Line 204: “It has been identified…": Replace by “No task x group interaction effect was observed for...”
Best regards
Your reviewer
Author Response
Response to the Reviewer: 1
We would like to thank the Reviewer for constructive and competent criticism. The manuscript has been revised according to the comments raised by the reviewer to the best of our ability. Changes to the manuscript are featured and highlighted in red color. Please find a detailed reply to the comments attached to this revision.
- Comments to the Author
Major issue:
One meter of height above ground (line 123) does not seem to represent an adequate level of apparent threat as “falling” down this distance (on gymnastics mattresses) is unlikely to produce any kind of discomfort to the subjects. It is beyond question, of course, health and safety of participants must be guaranteed throughout any study. However, increasing the height to. e. g., 1.5 m, or even 2 m, when adequately protected by mattresses, would have been equally harmless for healthy adult men. Results for those heights would have been more revealing for the outcome of the study, as you discuss yourselves in lines 298 and 314–317. So, why did you choose only 1 m of height?
Response:
We thank the Reviewer for this important suggestion. We are aware that 1 m height may not be dangerous enough to cause sufficient individual variability in the HW group and we mentioned it in our limitation and also in discussion.
According to OHS, work at height occurs when an employee works on a surface located more than 1 meter from the floor level and the place is uncovered and unsecured with permanent structures that will prevent falling and falling from heights. So if an employee works on a device, machine, or scaffolding that is more than 1 m from the floor, it is work at height. If the site is secured in one of the ways described above, it is not work at height. In our research, the platform where the respondent stands is uncovered and unsecured with permanent structures that will prevent falling and falling from heights that why we decided that 1 m height should be sufficient.
- Comments to the Author
Lines 103–104: “When changing altitude, greater cardiovascular stress occurs in office workers.” Please clarify if that “greater” refers to the intra-group (OW) or inter-group comparison. What was your corresponding assumption for HW?
Response:
Lines 103–104: The phrase “greater” refers to the inter-group comparison. We clarified the text “than in high-altitude workers” (current lines 115-116).
- Comments to the Author
Line 109: What was your precise definition of OW? E.g., starting at what average numbers of hours seated per business day?
Response:
Line 109: In our research, we did not establish a minimum number of hours for office workers. The group consisted mainly of administrative personnel. As far as we know, study participants were responsible mainly for providing various kinds of administrative assistance in the office. They were also responsible for the appropriate flow of information between the various departments of their company, as well as contact with the company's customers and contractors. They worked mainly in a sitting position in front of the computer. We highlighted it in lines 13 and 120-121.
- Comments to the Author
Line 113: Please state whether or not your research was conducted in accordance with the declaration of Helsinki.
Response:
Line 113: The information of accordance with the declaration of Helsinki has been added (current line 127).
- Comments to the Author
Line 245: "Overall level of physical activity”: According to your group comparisons, PA as evaluated with your tracking device before your experimental trials was *comparable* for both groups. So how can you implicitly suggest that PA was higher for the HW group as compared to the OW group? – Even though this assumption does seem conceivable to me as well. However, you have to stick to your data or explain why your data may not be sufficient to provide a complete picture.
Response:
Thank to Reviewer for this remark. Indeed, PA was comparable for both groups. However, in previous studies (DOI: 10.1177/1557988318774996), we have shown that the HW group had a higher rate of average physical activity at work than the OW group, whereas the OW group showed greater physical activity during leisure time.
Physical activity at work involves “training” the appropriate muscle group used during the performed task. We suspected that increased levels of physical activity at work (which is specific in at-hight work) may affect levels of stability.
- Comments to the Author
Language issues:
- Line 126: “Each trial *lasted* 30 seconds…” (erase “was”).
- Line 129: is *an* appropriate time (replace “the”)
- Line 133: “They kept…” (erase left)
- Lines 161, 182 and thereafter: mediolateral (replace “medial-lateral”)
- Line 164: “for *the* first minute” (erase one)
- Line: 181 “It has been reported…”: Replace by “There was no…”
- Line 183 and thereafter: Please set eta^2 in formulae style (2 as superscript)
- Line 204: “It has been identified…": Replace by “No task x group interaction effect was observed for...”
Response:
We thank the Reviewer for suggestions. Changes to the manuscript are featured and highlighted in red as follows.
- Line 126: The sentence has been changed (current line 139).
- Line 129: the vs. an (current line 142)
- Line 133: left vs. kept (current line 146)
- Line: 181 “It has been reported…” “There was no…” (current line 194).
- Line 183 and thereafter: eta^2 has been corrected in formula style (2 as superscript) in all text.
- Line 204: “It has been identified…" has been replaced by “No task x group interaction effect was observed for...” (current line 216).
Reviewer 2 Report
Comments on manuscript Number ijerph-920337 “The Effects of Cognitive Task and Change of the Heights on the Postural Stability and Cardiovascular Stress in Workers Working at Height.” The paper is well written and interesting. It shows clarity, coherence, and smooth flow of information. The main contribution of the authors is clearly presented. It only needs a minor revision before being accepted for publication. My comments are as follows:
The main conclusion presented in the abstract is about OW while the title of the manuscript HW as the main topic of the research
In line 19, it is unclear who or what This refers to.
In line 34, use “This unawareness” instead of “This”
In line 39 erase the comma that appears between “Poon” and “[9]”
In line 40, insert a comma after the word “organizational”
The first author have published some previous works (DOI: 10.1177/1557988318774996; DOI: 10.3390/s20133731) related to the topic. Please mention the different between her previous researchers and current work.
The references must be updated; in the introduction the authors must include research works of the current year (2020) in the literature review. No references of the current year are used in the introduction, please cite some relevant and recent references.
In line 149 define the acronym PA before use it.
In line 143 specify the type of instrument used the measure heart rate.
In line 241 reduce the space between the words “lower” and “results”
The authors should highlight findings that differ from findings in previous publications, and unexpected findings. Are your results consistent with what other investigators have reported? Or are there any differences? Why?
Conclusions are not meaningful, the authors do not show quantitative results. The conclusions should answer the aims of the study
Author Response
Response to the Reviewer: 2
We would like to thank the Reviewer for constructive and competent criticism. The manuscript has been revised according to the comments raised by the reviewer to the best of our ability. Changes to the manuscript are featured and highlighted in green color. Please find a detailed reply to the comments attached to this revision.
- Comments to the Author
The main conclusion presented in the abstract is about OW while the title of the manuscript HW as the main topic of the research.
Response:
Thank the Reviewer for this remark. According to your suggestion, the conclusion presented in the abstract has been changed (current lines 20-21).
- Comments to the Author
In line 19, it is unclear who or what This refers to.
Response:
The explanation has been added (current lines 20-21)
- Comments to the Author
In line 34, use “This unawareness” instead of “This”
Response:
It has been corrected (current line 35).
- Comments to the Author
In line 39 erase the comma that appears between “Poon” and “[9]”
Response:
Coma has been removed (current line 40).
- Comments to the Author
In line 40, insert a comma after the word “organizational”
Response:
Coma has been added (current line 41).
- Comments to the Author
The first author have published some previous works (DOI: 10.1177/1557988318774996; DOI: 10.3390/s20133731) related to the topic. Please mention the different between her previous researchers and current work.
Response:
The two previous publications and the current publication constitute one project for the implementation of a doctoral dissertation. The present work was the second in the series of articles, however it was submitted early to another journal, where despite positive reviews it was rejected. This resulted in the third work being published at an earlier date. A reference to the first work has been added in the introduction (DOI: 10.1177/1557988318774996) (current lines 96-100).
In the first work, we analyzed the level of postural stability and physical activity of at-height workers. Functional tests at the ground level were used to evaluate postural stability. In current work, we compare similar groups but in a more complex manner with the use of posturography in various conditions (eyes open and closed, cognitive task, low- height threat). In the third work, we assess the impact of proprioceptive training with the use of virtual reality (VR) on the level of postural stability of high–altitude workers. It was an experimental study with intervention.
- Comments to the Author
The references must be updated; in the introduction the authors must include research works of the current year (2020) in the literature review. No references of the current year are used in the introduction, please cite some relevant and recent references.
Response:
Thank the Reviewer for this remark. The new research works of the current year (2020) in the literature review have been added in the introduction and references.
- Chander, H.; Shojaei, A.; Deb, S.; Kodithuwakku Arachchige, S.N.K.; Hudson, C.; Knight, A.C.; Carruth, D.W. Impact of Virtual Reality–Generated Construction Environments at Different Heights on Postural Stability and Fall Risk. Workplace health & safety, 2020, 19, 2165079920934000, doi: 10.1177/2165079920934000 (current lines 66-70).
- Chinda, T.; Pongsayaporn, P. Relationships among factors affecting construction safety equipment selection: structural equation modelling approach. Civil Engineering and Environmental Systems, 2020, 37(3), 1-20, doi:10.1080/10286608.2020.1729754 (current line 32).
- Habibnezhada, M.; Puckett, J.; Jebelli, H.; Karji, A.; Fardhosseini, M.S.; Asadi, S. Neurophysiological testing for assessing construction workers' task performance at virtual height. Automation in Construction, 2020, 113, 103143, doi:10.1016/j.autcon.2020.103143 (current line 51).
- Sawicki, M.; Szóstak, M. Quantitative Assessment of the State of Threat of Working on Construction Scaffolding. International Journal of Environmental Research and Public Health, 2020, 17(16), 5773, doi: 10.3390/ijerph17165773 (current line 49).
- Comments to the Author
- In line 149 define the acronym PA before use it.
- In line 143 specify the type of instrument used the measure heart rate.
- In line 241 reduce the space between the words “lower” and “results”
Response:
We thank the Reviewer for these important suggestions. Changes to the manuscript are featured and highlighted in green as follows.
- In line 149- Define of the acronym PA has been added (current line 161).
- In line 143- the type of instrument used the measure heart rate has been added (current line 179).
- In line 241 - space between the words “lower” and “results” has been reduced (current line 257).
- Comments to the Author
The authors should highlight findings that differ from findings in previous publications, and unexpected findings. Are your results consistent with what other investigators have reported? Or are there any differences? Why?
Response:
It is difficult to answer this question because there are no previous studies comparing at-height workers with office workers. However, the overall view of the results related to changes in stability with the height of the platform and cognitive tasks is consistent with other studies where at height workers were examined in other contexts. We highlighted it in lines 250-253.
- Comments to the Author
Conclusions are not meaningful, the authors do not show quantitative results. The conclusions should answer the aims of the study
Response:
The conclusion has been corrected and corresponds to the hypotheses posed (current lines 336-338).
Reviewer 3 Report
Review of: The Effects of Cognitive Task and Change of the Heights on the Postural Stability and
Cardiovascular Stress in Workers Working at Height
By Dr. Evan A Nadhim
Objective: [13-15] The objective measures of the postural stability and cardiovascular stress were
evaluated for both groups at two different platform heights with or without cognitive task. “Can
you mention your evaluation methods?”
Justifications:
1. [61-62] They reported that by increasing the altitude of the surface on which the studied
person stands, the risk of fall perception goes up. [64-65] It was also identified that the
perception of loss of balance is reflected in the control of body balance as it alter the
strategy adopted by the central nervous system. “Any related studies to construction
industry”
2. [65-67] the effect of elevated height and lack of safety handrails increases anxiety and
subjective stress triggered by fear of falling, which has been demonstrated to affect the
neuromuscular system in both healthy and unhealthy people. “Any further references to
fall accidents”
3. Johnson et al., who researched emotional, cognitive and postural adaptations in young and
older adults under the influence of repeated exposure to danger, noticed that some
modifications in threat-related postural control may be closely correlated with emotional
response to the threat, while others may be dependent on the context. “Can you strengthen
your evidence with reference”
4. [91-92] One of these strategies may be to link the threat to more conscious control of body
balance. “The authors should have high certainty to link all mentioned elements”
Results: If it possible to include the results in tables in additions to the Figures.
Discussion: Some comments might improve this section:
1. [241-242] For all the analyzed parameters lower results (which may indicate a higher level
of postural stability) have been achieved by the HW group. “It logically to find HW group
has higher level of postural stability because you chosen 16 healthy men working at
height with one year’s experience of working at height. So what is the new
knowledge?”
2. [254-257] is there any suggestions?
3. [280-284] is it important to use the harnesses at such situations?
4. [290-299] Can you compare/link your results to the literature mentioned?
Finally: Dear authors, I really enjoyed reviewing you precise research.
God look in publishing this vulnerable work

Author Response
Response to the Reviewer: 3
We would like to thank the Reviewer for constructive and competent criticism. The manuscript has been revised according to the comments raised by the reviewer to the best of our ability. Changes to the manuscript are featured and highlighted in blue color. Please find a detailed reply to the comments attached to this revision.
- Comments to the Author
Objective: [13-15] The objective measures of the postural stability and cardiovascular stress were evaluated for both groups at two different platform heights with or without a cognitive task. “Can you mention your evaluation methods?”
Response:
Evaluation methods have been added (current lines 13-16).
- Comments to the Author:
[61-62] They reported that by increasing the altitude of the surface on which the studied person stands, the risk of fall perception goes up. [64-65] It was also identified that the perception of loss of balance is reflected in the control of body balance as it alter the strategy adopted by the central nervous system. “Any related studies to construction industry”
Response:
The entire section on current lines 61-66 is relevant to studies by Min, Kim and Parnianpour. Research concerns the effects of safety handrails and the heights of scaffolds on the subjective and objective evaluation of postural stability and cardiovascular stress in novice and expert construction workers.
- Comments to the Author:
[65-67] the effect of elevated height and lack of safety handrails increases anxiety and subjective stress triggered by fear of falling, which has been demonstrated to affect the neuromuscular system in both healthy and unhealthy people. “Any further references to fall accidents”
Response:
Further references to fall accidents in a group of construction workers have been added (current lines 64-74).
- Comments to the Author:
Johnson et al., who researched emotional, cognitive and postural adaptations in young and older adults under the influence of repeated exposure to danger, noticed that some modifications in threat-related postural control may be closely correlated with emotional response to the threat, while others may be dependent on the context. “Can you strengthen your evidence with reference”
Response:
The further references have been added (current lines 85-86).
- Comments to the Author:
[91-92] One of these strategies may be to link the threat to more conscious control of body balance. “The authors should have high certainty to link all mentioned elements”
Response:
We thank the reviewer for this important suggestion. We changed our statement for more soft (current line 103-104).
On the other hand, there are studies that confirm that anxiety is associated with an attentional bias for threatening stimuli. The anxiety may modify the processing of various sensory data involved in balance control irrespective of the level of anxiety in the subjects. Research also suggests that in people repeatedly staying at height with repeated exposure to stressful conditions, it reduces the level of fear and improves postural stability. The habituation effect may lead to the people being able to cope with the stressful situation better.
- Comments to the Author:
Results: If it possible to include the results in tables in additions to the Figures.
Response:
In fact, we have a table prepared but during previous reviews, it was suggested to change tables to figures and we tried to add all the most important information connected to results (interaction effects and main effects) on figures. We will leave the final decisions to the Editor of the Journal.
- Comments to the Author:
[241-242] For all the analyzed parameters lower results (which may indicate a higher level of postural stability) have been achieved by the HW group. “It logically to find HW group has higher level of postural stability because you chose 16 healthy men working at height with one year’s experience of working at height. So what is the new knowledge?”
Response:
Thank the Reviewer for this remark. Both groups include a 16 healthy men in a similar age, with a similar level of BMI and physical activity. According to the above, the results suggest that the type and nature of the work performed may have an impact on the level of postural stability. In fact, the hypothesis and results are quite predictable but on the other hand, there were no previous studies comparing this kind group of workers.
Moreover, previous studies have focused mainly on the effect of specified conditions (single-double task) or a specified level of height (low-high threat). To our best knowledge, there have been no studies that would examine all these elements together.
- Comments to the Author:
[254-257] is there any suggestions?
Response:
The results of the cited studies indicate that lower platform height may be insufficient to induce changes in postural stability. We emphasize this in the limitation section (current lines 330-334).
- Comments to the Author:
[280-284] is it important to use the harnesses at such situations?
Response:
When working, for example, on a ladder at a height of more than 2 m above the external ground or the floor, the use of a safety harness is obligatory. In the case of work performed at a height of more than 1 m and below 2 m, there is no such obligation - the regulations only require the use of appropriate protective measures for such work. The use of a harness during measurement could cause additional uncontrolled factors. Even the weight (around 2-3 kg) of the harness may influence the results of the postural stability measurement but it is a great idea for further studies.
- Comments to the Author:
[290-299] Can you compare/link your results to the literature mentioned?
Response:
Our results have been compared with the mentioned literature and they are also discussed and described on lines 316-329.